# Single-Cell Analyses Offer Insights into the Different Remodeling Programs of Arteries and Veins

**DOI:** 10.3390/cells13100793

**Published:** 2024-05-07

**Authors:** Miguel G. Rojas, Simone Pereira-Simon, Zachary M. Zigmond, Javier Varona Santos, Mikael Perla, Nieves Santos Falcon, Filipe F. Stoyell-Conti, Alghidak Salama, Xiaofeng Yang, Xiaochun Long, Juan C. Duque, Loay H. Salman, Marwan Tabbara, Laisel Martinez, Roberto I. Vazquez-Padron

**Affiliations:** 1Department of Surgery, Leonard M. Miller School of Medicine, University of Miami, Miami, FL 33136, USA; mgr98@med.miami.edu (M.G.R.); spereira@med.miami.edu (S.P.-S.); jtv6@med.miami.edu (J.V.S.); asalama@med.miami.edu (A.S.);; 2Bruce W. Carter Veterans Affairs Medical Center, Miami, FL 33125, USA; zmz4@miami.edu; 3Lewis Katz School of Medicine, Temple University, Philadelphia, PA 19140, USA; 4Medical College of Georgia, Augusta University, Augusta, GA 30912, USA; 5Department of Medicine, Leonard M. Miller School of Medicine, University of Miami, Miami, FL 33136, USA; 6Division of Nephrology and Hypertension, Albany Medical College, Albany, NY 12208, USA

**Keywords:** artery, vein, smooth muscle cell, fibroblast, synthetic, myofibroblast, remodeling, single-cell RNA sequencing, proteomics

## Abstract

Arteries and veins develop different types of occlusive diseases and respond differently to injury. The biological reasons for this discrepancy are not well understood, which is a limiting factor for the development of vein-targeted therapies. This study contrasts human peripheral arteries and veins at the single-cell level, with a focus on cell populations with remodeling potential. Upper arm arteries (brachial) and veins (basilic/cephalic) from 30 organ donors were compared using a combination of bulk and single-cell RNA sequencing, proteomics, flow cytometry, and histology. The cellular atlases of six arteries and veins demonstrated a 7.8× higher proportion of contractile smooth muscle cells (SMCs) in arteries and a trend toward more modulated SMCs. In contrast, veins showed a higher abundance of endothelial cells, pericytes, and macrophages, as well as an increasing trend in fibroblasts. Activated fibroblasts had similar proportions in both types of vessels but with significant differences in gene expression. Modulated SMCs and activated fibroblasts were characterized by the upregulation of *MYH10*, *FN1*, *COL8A1*, and *ITGA10*. Activated fibroblasts also expressed *F2R*, *POSTN*, and *COMP* and were confirmed by F2R/CD90 flow cytometry. Activated fibroblasts from veins were the top producers of collagens among all fibroblast populations from both types of vessels. Venous fibroblasts were also highly angiogenic, proinflammatory, and hyper-responders to reactive oxygen species. Differences in wall structure further explain the significant contribution of fibroblast populations to remodeling in veins. Fibroblasts are almost exclusively located outside the external elastic lamina in arteries, while widely distributed throughout the venous wall. In line with the above, ECM-targeted proteomics confirmed a higher abundance of fibrillar collagens in veins vs. more basement ECM components in arteries. The distinct cellular compositions and transcriptional programs of reparative populations in arteries and veins may explain differences in acute and chronic wall remodeling between vessels. This information may be relevant for the development of antistenotic therapies.

## 1. Introduction

Arteries and veins differ in histological and cellular organization. Consequently, these vascular conduits have different biomechanical properties [1], develop different types of occlusive lesions, and respond differently to injury [2,3]. Unlike arteries, veins rarely develop atherosclerosis, even under the supra-arterial circulation after the creation of arteriovenous fistulas [4,5]. On the other hand, the patency of mammary and radial artery grafts is superior to that of the saphenous vein after coronary revascularization surgery [6]. Veins often show higher rates of restenosis and increased fibrotic scarring after balloon angioplasty or stent placement [3,7,8]. Despite clinical and biological evidence, there have been modest efforts to understand the unique cellular and molecular mechanisms responsible for vein remodeling. This gap in knowledge prevents us from improving outcomes of current treatments for venous pathologies and from developing innovative vein-targeted therapies.

While it is widely recognized that the different etiologies of stenotic lesions in arteries vs. veins play a significant role in the efficacy of treatments [9,10], most comparative pathophysiological studies have been limited to histology [9,11,12]. Compared with veins, arteries have a thicker media and better-defined elastic laminas as an adaptation to high blood pressure and pulsatile flow [13,14]. The thinner intima of arteries is also attributed to high wall shear stress, and it allows efficient endothelial cell (EC)–smooth muscle cell (SMC) communication through myoendothelial gap junctions [15]. In contrast, veins frequently have a thick intima and loosely organized medial SMCs interspersed among abundant extracellular matrix (ECM) [2,16]. Many veins lack clear structural boundaries between wall layers due to extensive fragmentation or lack of elastic laminas, making the vein a distensible vascular tissue with extraordinary compliance.

Importantly, there is insufficient knowledge about the healing response of human veins, including the cellular dynamics during the early and late phases of remodeling [3,17,18]. Arguably, the most referenced cell types in vascular remodeling are “synthetic” SMCs, myofibroblasts, and mesenchymal cells, which are derived from phenotypic modulation of contractile SMCs, activated fibroblasts, and transformed ECs, respectively [19,20,21]. A common feature of these cells is a proinflammatory, proliferative, and ECM-producing phenotype that allows them to act as professional repair cells to restore hemostasis, scavenge dead cells and lipids, support neovascularization, and organize the ECM in response to acute and chronic stimuli [22,23,24]. Under dysregulated conditions, these same cells contribute to atherosclerosis, aneurysms, stenosis, and chronic venous diseases [22,23,25,26].

In the era of single-cell analysis, we have a unique opportunity to address gaps in knowledge about the processes of arterial vs. venous remodeling. With this in mind, this work compares human peripheral arteries (brachial) and veins (basilic/cephalic) by single-cell and bulk RNA sequencing, proteomics, and histology, with a focus on vascular cells with repair capacity (ECs, contractile and modulated SMCs, quiescent and activated fibroblasts). Our findings highlight the unique cellular and ECM environments in both types of vessels, as well as significant differences in the phenotypes of modulated SMCs and activated fibroblasts between arteries and veins that may underlie their unique remodeling tendencies. These new insights may have important implications for the design of arterial vs. venous therapies.

## 2. Materials and Methods

### 2.1. Study Subjects and Sample Collection

This study’s cohort included 30 deceased organ donors through a collaboration with the Life Alliance Organ Recovery Agency (no informed consent required; Appendix A). Cross-sectional samples ~2 cm in length from upper extremity vessels (18 basilic veins, 3 cephalic veins, 24 brachial arteries) were obtained following organ procurement procedures. Details about sample collection and storage are provided in Appendix A. This research did not require a determination from the Office of Human Subjects Research Protection or IRB approval.

### 2.2. Single-Cell RNA Sequencing

Single-cell suspensions of 3 brachial arteries and 3 veins (2 basilic, 1 cephalic) from 6 donors were obtained by enzymatic digestion (Appendix A). Single-cell RNA sequencing (scRNA-seq) was performed in the Center for Genome Technology at the University of Miami (UM) John P. Hussman Institute for Human Genomics. Alignment of raw sequencing data, quality control (QC), and bioinformatic analyses were performed as described in Appendix A. Sequencing data were deposited in the Gene Expression Omnibus (GEO), accession numbers GSE266682 and GSE250469.

### 2.3. Bulk RNA Sequencing

Total RNA was isolated from 7 pairs of arteries (brachial) and veins (6 basilic, 1 cephalic) from 7 independent donors as previously described [27]. Bulk RNA sequencing was performed at the UM John P. Hussman Institute for Human Genomics. Differential gene expression analyses and digital cytometry were performed as described in Appendix A. Bulk sequencing data were deposited under GEO accession numbers GSE266681 and GSE233264.

### 2.4. Protein Extraction and ECM-Targeted Proteomics

Proteins from 6 pairs of arteries and veins from 6 donors were extracted using a subfractionation method previously described with minor modifications (Appendix A) [28]. The ECM-enriched fraction of 3 pairs was profiled by LC-MS/MS at MS Bioworks (Ann Arbor, MI, USA). The soluble fraction of all samples was analyzed by Western blot (WB).

### 2.5. Validation Experiments

In vitro validation of differentially expressed genes (DEGs) or cell populations of interest was performed using WB, flow cytometry, histology, immunohistochemistry (IHC), or immunofluorescence (IF), as described in detail in Appendix A.

### 2.6. Statistical Analysis

Statistical analyses of nonsequencing data were performed using GraphPad Prism 8.4.0 (San Diego, CA, USA). Normally distributed data were compared using *t*-tests and expressed as mean ± standard deviation (SD) or mean ± standard error of the mean (SEM) where indicated. If normality assumptions were not met, the Mann–Whitney test was used, and data were expressed as median and interquartile range (IQR). Comparisons of paired arteries and veins from the same donors were performed using paired *t*-tests or paired Wilcoxon signed-rank tests as appropriate.

## 3. Results

### 3.1. Histology Highlights Structural Differences between Upper Arm Arteries and Veins

To discern biological differences between arteries and veins, we obtained seven pairs of brachial arteries and nearby veins (six basilic, one cephalic) from organ donors (Appendix A). One fragment of the tissue was formalin-fixed for histology, and another was used for total RNA isolation. Donors were 41 ± 19 years old (mean ± standard deviation) and had a similar representation of sexes and diverse racial and ethnic characteristics.

Brachial arteries demonstrate a well-organized media with tightly packed SMCs, clearly defined elastic laminas, and a negligible intimal layer (Figure 1A). In contrast, the walls of basilic veins were less muscular (31.9 vs. 57.1%, *p* = 0.0002) and had fragmented elastic laminas. Venous SMC formed bundles surrounded by abundant ECM showing a less restricted organization (Figure 1B,C). Veins also had a thicker intimal layer compared with nearby arteries (54.9 vs. 23.4 μm, *p* = 0.023) (Figure 1C). Of note, due to the deep anatomical location of basilic veins, the increased intimal thickness is likely a histological feature and not related to cannulation injury [16].

### 3.2. Single-Cell RNA Sequencing Contrasts the Arterial and Venous Cellular Environments

For a granular dissection of cell composition in both types of vessels, brachial arteries (ages 28–59, one female and two males) and upper arm veins (two basilic, one cephalic; ages 26–70, two females and one male) from six independent donors were profiled by scRNA-seq. Figure 1D presents the uniform manifold approximation and projection (UMAP) plots of arterial and venous cells after QC filters. The cellular atlas confirmed a 7.8-fold higher abundance of contractile SMCs (top Seurat markers: *ACTA2*, *TAGLN*, *MYH11*, *CNN1*) in arteries, and higher percentages of ECs (*VWF*, *PECAM1*, *CLDN5*, *ACKR1*; four-fold), monocyte/macrophages (*CD163*, *IL1B*, *CCL3*, *CD14*; 3.1-fold), and pericytes in veins (*STEAP4*, *FABP4*, *ACTA2*, *TAGLN*; 4.8-fold) (Figure 1E, Table 1). Modulated SMCs, defined by lower expression of contractile markers and upregulation of *MYH10*, *FN1*, and *COL8A1* with respect to contractile SMCs (Appendix A), showed a trend toward higher abundance in arteries, whereas fibroblasts had an increasing trend in veins (*DCN*, *PDGFRA*, *CFD*, C3; Table 1). Activated fibroblasts were characterized by significant upregulation of *COL1A1* and *LUM* with respect to the latter and negligible expression of *CFD* and *C3* (Appendix A). These were found at similar proportions in both types of vessels. Similarly, there were no differences in the proportions of NK/T cells (Table 1).

### 3.3. Cell Composition Differences Are Validated by Digital Cytometry

Single-cell gene expression signatures for the four main vascular populations (ECs, SMCs, fibroblasts, and immune cells) were used to deconvolute the bulk RNA sequencing profiles of the seven artery–vein pairs analyzed by histology (Appendix A). Modulated SMCs and activated fibroblasts were not included in this analysis due to their intermediate transcriptional profiles and a small number of unique markers. Compared with cell proportions detected by scRNA-seq analyses, the deconvolution algorithm favored the detection of well-differentiated populations (SMCs, ECs, and immune cells), while underestimating the proportion of fibroblasts. Nonetheless, pairwise comparisons between arteries and veins validated the relative differences detected by scRNA-seq, including the lower proportion of SMCs and a higher percentage of ECs and fibroblasts in veins (Appendix A).

Pairwise differential gene expression analyses of bulk RNA profiles further corroborated the results from digital cytometry (Appendix A). Out of 4671 differentially expressed genes (DEGs) between arteries and veins (baseMean > 50, FDR < 0.05), 2200 were identified as enriched in SMCs, ECs, fibroblasts, or immune cells by scRNA-seq. A heatmap of the latter subset demonstrates that genes upregulated in arteries are predominantly SMC-derived, while those upregulated in veins are enriched in ECs, immune cells, and fibroblasts (Appendix A).

### 3.4. Unique Adhesive and Hemostatic Properties of Arterial and Venous EC Populations

Vascular ECs comprise a wide diversity of transcriptional phenotypes depending on their physiological states and macro- or microvascular locations [29]. Combined with hemodynamic conditions, this phenotypic variability influences vascular function and remodeling processes. In addition to the above differences in EC abundance between arteries and veins, the identification of EC phenotypes by subclustering analysis revealed a striking contrast in vessel-specific subpopulations (Figure 2A). Six main EC phenotypes were uncovered by this analysis: (1) *ACKR1*+, previously identified in the main lumen and venules of veins [29]; (2) *SEMA3G*+, of arteriolar or capillary characteristics; (3) *ITLN1*+, here identified as the predominant phenotype in brachial arteries; (4) *EFEMP1*+, valvular-like ECs; (5) *PROX1*+, lymphatic ECs; and (6) *AIF1*+, an inflammatory subtype also detected in this study. *ITLN1*+ cells represent more than half of arterial ECs and are barely detected in veins (Figure 2A, Table 2). These were found lining the main lumen of arteries by IHC (Figure 2B). The minor *ACKR1*+ and *SEMA3G*+ populations in arteries are likely derived from adventitial microvasculature. In contrast, veins present a higher diversity of EC subtypes, including a higher abundance of *AIF1*+, *SEMA3G*+, and *EFEMP1*+ cells (Table 2).

Interestingly, macrovascular populations in arteries and veins express different arrays of leukocyte adhesion receptors. Venous *ACKR1*+ ECs show upregulation of *SELE*, *VCAM1*, and *ICAM1*, while *ITLN1*+ arterial cells have a higher expression of *SELL*, *SELP*, and *NCAM1* (Figure 2C, Appendix A). The inflammatory *AIF1*+ subtype is characterized by the upregulation of *CD44*, *MARCO*, *F13A1*, and *C5AR1*. However, it shares markers in common with *ACKR1*+ and *SEMA3G*+ cells (Figure 2C), which suggests that it represents a transcriptional state of the latter populations. Arterial and venous ECs also demonstrate different hemostatic characteristics. While *ITLN1*+ cells show upregulation of complement cascade components (*C1R*, *C1S*, *CFH*), serpin family genes (*SERPINE1*, *SERPING1*), and prostaglandin synthase genes (*PTGIS*, *PTGS1*, *PTGS2*), *ACKR1*+ ECs express inhibitors of NO signaling (*NOSTRIN*), the coagulation factor *F8*, and tissue-type plasminogen activator (*PLAT*) (Figure 2D). These differences reveal distinct regulatory mechanisms of the blood coagulation cascade in arteries and veins.

The increased diversity of EC subtypes in veins and the overall higher proportion of ECs compared with arteries can be explained by the higher counts of vasa vasorum in the venous wall (Figure 2E,F). This may also be related to the higher proportion of pericytes in veins interacting with microvessels and elevated counts of infiltrated monocyte/macrophages [29] (Table 1). Table 3 presents a summary of markers for the main EC subtypes in arteries and veins.

### 3.5. Arterial and Venous SMC Profiles Reflect Distinct Hemodynamic Environments

Smooth muscle cells play an essential role in vascular development and are one of the main cell types involved in arterial wall remodeling [23]. In arteries and veins, these cells express canonical contractile markers such as *ACTA2*, *MYH11*, *TAGLN*, and *CNN1* (Figure 3 and Appendix A). However, in addition to the pronounced contrasts in SMC proportions, Figure 3A demonstrates significant transcriptional differences between arterial and venous SMCs (Appendix A). Examples of DEGs include the upregulation of *COL8A1* and *IGFBP2* in arteries, which were confirmed by IHC or WB (Figure 3B,C).

Functional scores were developed based on curated gene signature modules from the Molecular Signatures Database (MSigDB) [30,31] and genes associated with phenotypic modulation of SMCs in atherosclerosis and aneurysmal disease [32,33]. Interestingly, DEGs in SMCs between both types of vessels are reminiscent of functional differences and/or hemodynamic adaptations (Figure 3D). Arterial SMCs have higher contractility, phenotypic modulation, and cell-to-wall adhesion scores than their venous counterparts (Figure 3D). In contrast, venous SMCs have higher scores of inflammation and adaptation to heme and low-oxygen blood.

The difference in contractility scores between vessels is mainly due to the upregulation of cytoskeleton-associated proteins in arteries (*CNN1*, *SORBS1*, *VCL*, *ITGA1*) and vasodilatory guanylate cyclase subunits in venous SMCs (*GUCY1A1*, *GUCY1B1*). In contrast, core contractile genes such as myosin chains and kinases (e.g., *MYH11*, *MYL9*, or *MYLK*) are not differentially expressed (Figure 3D, Appendix A). Genes upregulated in arteries and predictors of prosynthetic modulation include the transcription factor *KLF4*; secretable factors *CCN2*, *IGFBP2*, and *TNFRSF11B*; and prosynthetic ECM components such as *COL8A1*, *FN1*, and *FMOD* [34,35,36] (Figure 3D, Appendix A). Along with *FN1*, upregulation of the fibronectin receptors *SDC4* and *ITGA5* in contractile SMCs from arteries supports a role for this ECM component in the modulation of the SMC phenotype. Table 4 presents a summary of markers and gene expression differences between arteries and veins.

Peripheral veins return blood that is low in oxygen and rich in waste byproducts back to the heart. Over time, this unique hemodynamic environment may underlie adaptive and/or reactive transcriptional changes in venous SMCs. Inflammatory genes upregulated in veins include *S100A4*, *CCL2*, *CXCL12*, *PGF*, and *VEGFA* (Figure 3D). The latter may also represent a proangiogenic response to low oxygen tension. The upregulation of *COX4I1* and *NDUFA4L2* in venous SMCs likely reflects an optimization of the electron transport chain in this low-oxygen environment [37,38], while increased *HMOX2* generates carbon monoxide from heme group degradation and provides an alternative vasodilatory substrate to activate guanylyl cyclase [39].

### 3.6. Inflammation and Response to Oxidative Stress Are a Common Theme in Venous Fibroblasts

Quiescent fibroblasts were identified by a high expression of *PDGFRA*, *DCN*, *LUM*, *CFD*, *C3*, and collagen genes in both types of vessels (Figure 4 and Appendix A). Like SMCs, arterial and venous fibroblasts are different at the transcriptional level. Figure 4A presents the top DEGs between arterial and venous fibroblasts while demonstrating significant phenotypic heterogeneity within each type of vessel (Appendix A). The structural differences between arteries and veins also affect the spatial distribution of fibroblasts. The external elastic lamina of arteries keeps most PDGFRA^+^ fibroblasts in the adventitia, while they are widely distributed throughout the venous wall (Figure 4B).

Gene signature scores indicated functional differences in ECM deposition, inflammation, and oxidative stress responses between arterial and venous fibroblasts (Figure 4C). While collagen expression scores were similar between vessels, arterial fibroblasts had higher RNA levels of collagen-binding proteoglycans, including *PRELP*, *OMD*, *OGN*, *LUM*, and *ASPN* (Table 4). Differential expression of these proteoglycans may be relevant to vascular diseases, given their regulatory role in collagen fibril formation, extracellular biomineralization, and osteogenic cell differentiation [40,41,42,43,44,45]. Interestingly, as in SMCs, venous fibroblasts seem better responders to low oxygen tension. These show upregulation of *HIF1A* and its post-translational regulator *EGLN1* [46], the hypoxia-inducible genes *PSMB3* and *UBC* [47], as well as *VEGFA* (Figure 4C).

Genes involved in the detoxification of reactive oxygen species (ROS) are upregulated in venous fibroblasts compared with their arterial counterparts (Figure 4C, Table 4). These include major regulators of redox homeostasis in mammalian cells, such as *TXNRD1*, *P4HB*, *PRDX6*, and *SOD2* [48]. Lastly, gene signature scores suggest a higher inflammatory status in venous vs. arterial fibroblasts. Proinflammatory genes with higher expression in veins include *MIF*, *EGFR*, *IL1R1*, and chemokines such as *CXCL2*, *3*, and *12*. To balance these inflammatory signals, venous fibroblasts have higher expression of adrenomedullin (*ADM*), an important hormone peptide with anti-inflammatory and vasodilatory functions [49] (Figure 3C, Appendix A).

### 3.7. Phenotypic Differences in Myofibroblastic Populations Explain the Fibrotic Tendency of Veins

The differences in relative proportions of myofibroblastic cell types between arteries and veins piqued our interest due to their widely accepted role in vascular remodeling. Specifically, there is a balanced proportion of modulated SMCs and activated fibroblasts in arteries (8 vs. 10.6%, respectively; Table 1), but a 4.5X higher fraction of activated fibroblasts vs. modulated SMCs in veins (10.3 vs. 2.3%). This may be related to the tendency of veins for fibrotic remodeling with age and acutely after vascular surgeries [16,27,50]. Also referred to as “synthetic” SMCs, myofibroblasts, or fibromyocytes [32,51,52], phenotypically modulated mural cells are identified by coexpression of contractile (e.g., *ACTA2*, *MYH11*, *CNN1*, *TAGLN*) and fibroblast markers (e.g., *LUM*, *DCN*, *PDGFRA*, *COL1A1*) (Appendix A). The terms “modulated SMC” and “activated fibroblast” denote a presumed closer relationship to SMC and fibroblasts, respectively, which may be related to putative cellular origins or coexisting phenotypic states along SMC- or fibroblast-specific transcriptional spectra (Figure 5).

Modulated SMCs express SMC genes, such as *MYH11*, *CNN1*, and *ITGA8*, but show significant upregulation of “synthetic” markers like *MYH10*, *FN1*, and *COL8A1* (Figure 5B, Appendix A). Proinflammatory adhesion molecules (*ALCAM*) [53], promigratory mediators (*PCDH7*) [54], and modulation regulators (*SPINT2*) [55] are also upregulated in this subcluster. High levels of *MYH10*, *FN1*, *COL8A1*, and *POSTN* are also defining features of activated fibroblasts, along with significant upregulation of the thrombin receptor *F2R*, and the integrin *ITGA10* (shared by arterial modulated SMCs). The profibrotic marker *COMP*, which catalyzes collagen fibril formation, and *GPM6B*, an activator of TGFβ-SMAD2/3 signaling [56], are additional characteristics of this subcluster (Figure 5B, Appendix A).

Validation of activated fibroblasts in arteries and veins was performed by flow cytometry based on the coexpression of *F2R* and the fibroblast marker *THY1* (CD90) (Appendix A). The percentages of double-positive cells were in line with those detected by scRNA-seq. Similarly, the marker *ITGA10* was selected for histological validation. The morphology of ITGA10^+^ cells in arteries and veins reflects the possible different origins of this subset of myofibroblastic cells in arteries and veins (Appendix A). Furthermore, the generalizability of markers for modulated SMCs and activated fibroblasts was also confirmed in 3 published [57,58] coronary arteries and 3 saphenous veins (Appendix A).

Interestingly, gene signature scores associated with vascular remodeling processes indicate significant functional differences among myofibroblastic subclusters between arteries and veins (Figure 5C). Activated fibroblasts from veins are the top producers of collagens compared with quiescent fibroblasts from the same type of vessel and both kinds of arterial fibroblasts (Figure 5B,C, Appendix A). Accordingly, the biological processes of ECM organization and collagen biosynthesis, fibril organization, and metabolism show higher overrepresentation in activated fibroblasts from veins (Figure 6A). The processes of collagen-activated receptor tyrosine kinase signaling and inflammatory response to wounding are also uniquely enriched in this subcluster. Specific genes supporting these pathways include a 4 to 11-fold upregulation of collagens type I, III, and VI in veins vs. arteries (Figure 6B, Appendix A). *PCOLCE* and *COMP*, which boost collagen precursor maturation and fibrillation, and *TIMP1*, a critical mediator of remodeling, show similar levels of upregulation. Cell chemotaxis was also found over-represented in activated fibroblasts and quiescent fibroblasts from veins. This is supported by ECM remodeling factors (*PPIB*, *LOX*) and mediators of cell migration and chemotaxis (*PDGFRA*, *MIF*, *LGALS3*, *RAB13*), which are upregulated 1.6- to 2.4-fold in these venous populations.

Collagen production, inflammation, and oxidative stress are intricately associated mechanisms in fibrotic remodeling. Functional scores and gene ontology (GO) over-representation analyses highlight the participation of activated and quiescent fibroblasts from veins in cytokine production, immune cell infiltration, and responses to reactive oxygen species (ROS; Figure 5C and Figure 6A). Of note, the over-representation of responses to mechanical stimulus is uniquely identified in venous fibroblasts (Figure 6A). This suggests these quiescent fibroblasts are a continuous source of the activated population in the setting of vascular injury. Similar to activated fibroblasts, modulated SMCs in veins also show upregulation of collagens (Figure 6C, Appendix A), further demonstrating the profibrotic character of venous repair. In addition, modulated SMCs in veins seem to have unique metabolic adaptations as demonstrated by the over-representation of glycolysis and ADP metabolic biological processes (Figure 6A).

While upregulation of ECM components is a defining characteristic of myofibroblastic populations in veins, modulated SMCs and activated fibroblasts from arteries have significant upregulation of genes previously associated with atherosclerotic disease or aneurysm development. These include prosynthetic SMC genes such as *COL8A1*, *CCN2*, *IGFBP2*, *LTBP2*, and *TNFRSF11B* [32,33] (Table 4), atheroprotective factors (*RGS5*, *CCN3*, *CLU*) [59,60,61], and genes involved in vascular integrity and elastogenesis (*FBLN5*, *ELN*) (Figure 6B,C, Appendix A). Activated fibroblasts from arteries also show upregulation of genes involved in contractility (*MYH11*, *ACTA2*, *PLN*, *LMOD1*, *PPP1R14A*) compared with their venous counterparts. This similarity with SMCs was also observed in activated fibroblasts from published coronary arteries and contrasts the transcriptional profiles of activated fibroblasts from both basilic/cephalic (this study) and saphenous veins (Appendix A). The transcriptional differences between activated fibroblasts from arteries and veins may reflect distinct differentiation mechanisms in their respective vascular environments, or a poor resolution to fully separate SMC- and fibroblast-derived myofibroblastic subclusters.

### 3.8. Transcription Factors and Cell-to-ECM Interactions Contribute to Differences in Phenotypic Modulation

Transcriptional regulation and outside-in signaling from the ECM are major mechanisms responsible for controlling the vascular SMC phenotype and fibroblast activation [20,23,34]. To further investigate the mechanisms supporting SMC modulation in arteries and fibroblast differentiation in veins, we searched for differences in transcription factors (TFs) and ECM genes among vessels. From the 493 DEGs between arterial and venous contractile SMCs (absolute log_2_FC > 0.5, *p*-adjusted < 0.01), 34 are functionally classified as transcriptional regulators by the DAVID Bioinformatics database [62] (Appendix A). Eight of these TFs are associated with prosynthetic modulation of SMCs (*EGR1* [63], *FOS* [64], *JUN* [64], *KLF4* [65], *YAP1* [66], *FOXP1* [67], *FHL5* [68], *LMCD1* [69]) while nine have been shown to oppose modulation (*FOXC1* [70], *BASP1* [71], *TCF4* [72], *TBX2* [73], *PRRX1* [74], *SMARCD3* [75], *BTG1* [76], *WWTR1* [77], *SF1* [78]). Interestingly, all eight promodulation factors are upregulated in arteries, while 6/9 antimodulation transcriptional factors are upregulated in veins (Figure 7).

In contrast to the artery–vein dichotomy of TFs associated with SMC modulation, there seems to be a balance of pro- and antimodulation ECM components in the walls of both vessel types. Prosynthetic modulation of SMCs is promoted by fibronectin and collagen types I, III, VIII, and XVIII, while opposed by collagen type IV and laminin [34,35,36,79]. Major fibrillar collagens (types I, II, and III) also determine ECM stiffness, which activates fibroblast differentiation to myofibroblasts [20]. We combined contractile SMCs, modulated cells, activated fibroblasts, and fibroblasts to estimate the overall expression levels of ECM genes in the wall. Promodulation *FN1*, *COL8A1*, and *COL18A1* are upregulated in arteries (Figure 8A). In contrast, genes encoding for collagens I, III, and V (type V is essential for fibrillation of types I and III) [80] are upregulated in veins. Among antimodulation ECM proteins, collagen IV genes have higher expression in arteries, while laminin has higher RNA levels in veins. In terms of ECM receptors, expression of integrin subunits *ITGA1* and *ITGA5*, which recognize collagens and fibronectin, respectively [81], are upregulated in arterial vs. venous contractile SMCs, while the opposite pattern is observed for laminin binding *ITGA3* (Figure 8B). From the transcriptional regulation point of view, the above comparisons strongly suggest an intrinsic resistance of venous SMCs to acquire a synthetic phenotype. On the other hand, the origin of modulated SMCs and activated fibroblasts in veins appears intricately related to fibronectin and collagen deposition, as illustrated by upregulation of *ITGA1*, *ITGA2*, *ITGA5*, *ITGA10*, and *ITGA11* with respect to contractile SMCs and quiescent fibroblasts (Appendix A). This agrees with a process of fibroblast-to-myofibroblast transition in response to changes in stiffness [20].

The differences in the above extracellular proteins at the RNA level were further investigated by ECM-targeted proteomics. Mass spectrometry of ECM-enriched fractions from three pairs of brachial arteries and basilic veins detected a total of 1083 proteins, with 75% of these in more than one sample (Appendix A). Over 22% of the proteins corresponded to core ECM and associated cytoskeletal genes. These two groups also had the highest NSAF values, in agreement with the enrichment protocol. The most abundant collagens in arteries and veins were types I through VI and XIV. In addition, approximately half of the detected glycoproteins and proteoglycans were as abundant as the collagens above (Appendix A). From the 77 ECM proteins identified in at least two of the samples, 28 had a *p*-value ≤ 0.1 in paired *t*-tests between arteries and veins, and 12 were differentially expressed with log_2_FC > |1| and *p* < 0.05 (Figure 9). Importantly, proteomics analyses confirmed the differences in ECM composition between arteries and veins and the increased profibrotic nature of the latter.

### 3.9. Global Interactome Highlights Proangiogenic and Inflammatory Signals in Veins

We performed interactome analyses with the main cell clusters via CellChat to further compare cell-to-cell communication networks in arteries vs. veins. While several pathways were significantly enriched in arteries (e.g., GDFs, BMPs, periostin, FGF), in terms of signal strength and differences between vessels, the most predominant pathways were those enriched in veins. These included galectins, angiopoietin-like factors (ANGPTLs), chemokines (CCLs, CXCLs), and VEGFs (Appendix A). The main three angiopoietin-like factors differentially expressed between arteries and veins are *ANGPTL1*, *2*, and *4*. These three secretable factors are involved in the regulation of angiogenesis, while *ANGPTL4* is also involved in lipid metabolism, redox regulation, and inflammation [82]. All three are upregulated in venous modulated SMCs, activated fibroblasts, and fibroblasts, whereas *ANGPTL1* has higher expression levels in arterial contractile SMCs (Appendix A). Reinforcing proangiogenic signaling, *VEGFA* is upregulated in *EFEMP1*+ ECs, contractile SMCs, modulated SMCs, activated fibroblasts, and fibroblasts from veins compared with arteries, as are *FLT1* (VEGFR1) and *KDR* (VEGFR2) in *ACKR1*+, *SEMA3G*+, and *PROX1*+ ECs (Appendix A).

Venous fibroblasts are also high expressors of proinflammatory chemokines compared with arterial ones, including *CXCL2*, *3*, and *12*. Similarly, *CXCL12* shows higher levels in venous vs. arterial contractile SMCs, modulated SMCs, and activated fibroblasts, as is the case of *CCL21*, which is upregulated in venous *PROX1*+ ECs and contractile SMCs (Appendix A). Altogether, the above cell-to-cell communication networks demonstrate an active cellular ecosystem in veins that supports cell migration, intramural vascularization, and inflammation.

## 4. Discussion

It is often assumed that unique biological mechanisms govern repair and remodeling in arteries and veins without substantial scientific evidence. This assumption has led to the application of artery-developed treatments to treat venous occlusions with worse outcomes [2]. Using comparative bulk and single-cell RNA sequencing and proteomics of upper arm arteries and veins, we challenge this idea and reveal the distinct cellular and molecular characteristics of these vascular beds. From the arterial side, our study reveals (1) a high potential in SMCs for phenotypic modulation toward myofibroblastic cells even in the absence of disease; (2) the existence of an extracellular scaffold rich in fibronectin and collagen type VIII that could facilitate such modulation; and (3) a potential contribution of SMCs and fibroblasts to reparative myofibroblastic populations (modulated SMCs and activated fibroblasts). From the venous side, we demonstrate (1) metabolic adaptations of SMCs to hypoxia and heme; (2) predominantly fibroblast-derived reparative cells with proangiogenic and -fibrotic characteristics; and (3) an involvement of ECs, SMCs, myofibroblasts, and fibroblasts in proinflammatory signaling. This information will be crucial for a better understanding of cellular dynamics within the walls of arteries and veins and of mechanisms responsible for differential vascular remodeling after revascularization and vascular access surgeries.

Our unique comparative transcriptomic approach of peripheral arteries and veins allowed the identification of reparative myofibroblastic cells (modulated SMCs and activated fibroblasts) and their functional characteristics. Similar populations, named modulated or intermediate SMCs, have been previously identified in human aortas based on analyses of diseased tissues [32,52]. However, until now, the transcriptional profiles of phenotypically modulated mural cells and activated fibroblasts in healthy arteries and veins had remained elusive. These cells showed a primarily repair-driven transcriptional program with components that address hemostasis (*F2R*), ECM integrity and collagen deposition (*FN1*, *POSTN*, *COMP*, *COL8A1*,), and cell proliferation, migration, and differentiation (*FN1*, *POSTN*, *COMP*, *COL8A1*, *ITGA10*). The protease-activated thrombin receptor (*F2R*) is further upregulated after injury to promote coagulation, as well as SMC and fibroblast proliferation, migration, and ECM deposition [83]. Fibronectin is one the first ECM proteins produced in vessels during embryonic development, and along with its receptor α5β1 (also upregulated in modulated SMCs and activated fibroblasts), is critical for vascular morphogenesis [84,85]. Periostin and collagen type VIII are also induced after vascular injury to promote the proliferation and migration of myofibroblasts [86,87]. *ITGA10*, in turn, is a collagen-binding subunit with restricted cell expression that promotes cell survival and proliferation through increased PI3K/AKT signaling [88].

Interestingly, our data suggests that venous reparative cells originate mostly from fibroblasts while, in arteries, these cells are related to both SMCs and fibroblasts. The increased phenotypic modulation potential of arterial SMCs can be explained by their higher proportion in the wall compared with veins, the upregulation of prosynthetic TFs and ECM components, and a physiological requirement imposed by the structural constraints of the elastic laminas. In this well-delimited environment, homeostasis of the media must be performed by SMCs and to a lower extent migrating fibroblasts. This may pose a risk for atherosclerosis development where dysregulated phenotypic switching of SMC initiates atheroma formation and becomes a source for foam cells and plaque osteoblasts [23]. In contrast, the open structure of veins is conducive to fibroblast and myofibroblast migration. The widespread distribution of fibroblasts throughout the wall may also explain the high ECM content of the venous intima and media and agrees with the upregulation of fibrillar collagens validated by proteomics. This collagen-enriched environment also promotes continued fibroblast activation [20], a tendency of veins to develop fibrosis over time [16], and fibrotic remodeling after vascular access surgeries and endovascular treatments [50]. It is likely that activated fibroblasts in veins have limited plasticity towards osteoblastic and myeloid phenotypes, providing atheroresistance in these conduits.

There is enough circumstantial evidence in our cellular atlas to postulate that reparative myofibroblastic cells in healthy vessels can go awry and lead to diseases. Modulated SMCs in arteries had significant upregulation of proatherosclerotic genes such as *IGFBP2*, *TNFRSF11B*, and *CCN2*. Although less common than in coronary and carotid arteries, atherosclerosis in the brachial artery is a relatively frequent form of peripheral arterial disease [11]. The potential contribution of SMCs to the pool of cholesterol-rich cells in the atheroma has been suggested using animal models [89]. Therefore, the phenotypic information uncovered by our analyses may be of interest for the development of preventive therapies. In contrast to modulated SMCs and activated fibroblasts in arteries, activated fibroblasts from veins are the highest producer of collagens and proangiogenic signals. Along with increased detox activity of ROS and heme, angiogenesis in veins reflects an adaptive response to hypoxic conditions. This is exemplified by the upregulation of *NDUFA4L2* in venous contractile and modulated SMCs compared with arterial cells, as well as by the over-representation of glycolytic biological processes in the former. Dysregulated angiogenesis and fibrosis are some of the features of chronic venous insufficiency. Importantly, the above transcriptional and metabolic differences may underlie the maladaptive response of vein grafts to the arterial environment and contribute to the development of vein graft disease.

Of particular interest for the study is to find clues that explain the worse outcomes of veins compared with arteries after vascular injury. Several insights can be drawn from this study. First, having a reparative population that is derived from widespread fibroblasts makes it improbable that the vein will ever have a lasting response to plain, drug-eluting, or stent angioplasty. Second, killing any medial SMCs with these methods will most likely lead to overpopulation of the media with migrating activated fibroblasts. Third, an exacerbated fibrotic response to the injury will incite the activation of additional fibroblasts. Lastly, the increased inflammatory milieu in veins vs. arteries by virtue of chemokine-expressing ECs, SMCs, myofibroblastic cells, and fibroblasts is a primed environment for fibrosis, cell migration, and proliferation, as well as for exacerbated responses to synthetic materials, as in the case of venous-ePTFE graft anastomoses. The significant gene expression differences in leukocyte adhesion receptors and hemostatic mediators in arterial vs. venous ECs also point to vessel-selective mechanisms for leukocyte infiltration and a possibility to tailor antithrombotic therapies.

The limitations of this study include the small number of vessels analyzed by scRNA-seq and the potential contribution of donor-specific characteristics to gene expression differences.

## 5. Conclusions

This study presents a much-needed comparative analysis of peripheral arteries and veins at the bulk and single-cell gene expression, proteomics, and histological levels. We highlight unique transcriptional properties of ECs, mural cells, and fibroblast populations from both types of vessels, with profound implications for the study of vessel-specific acute and chronic remodeling processes. Based on the significant differences in cell proportions, phenotypes, and cell-to-cell interactomes identified, we urge the scientific community to take advantage of these publicly available gene expression resources and to stop once and for all extrapolations from arterial biology to explain vein biology.

## Figures and Tables

**Figure 1 cells-13-00793-f001:**
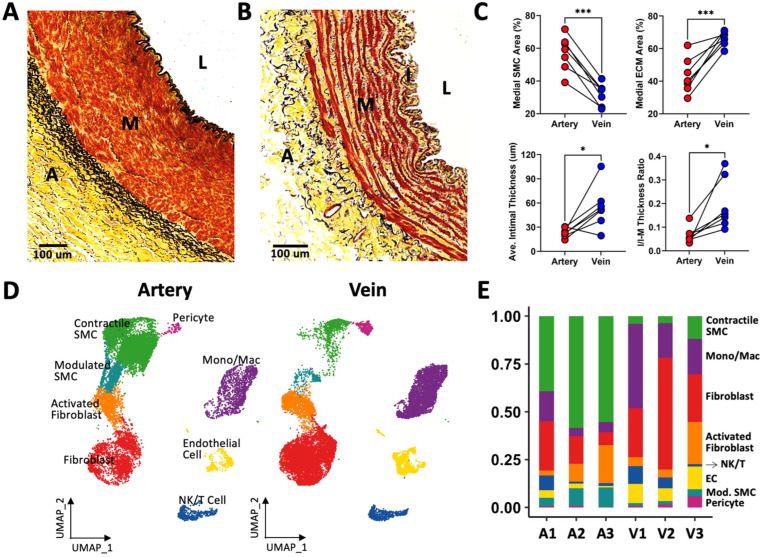
Differences in wall morphometry and cell composition between upper arm arteries and veins: (**A**,**B**) Representative Movat’s pentachrome stained cross-sections of a brachial artery (**A**) and a basilic vein (**B**), indicating the lumen (L), intima (I), media (M), and adventitia (A). Cells are stained in red, collagen in yellow, and elastin in black. (**C**) Morphometric comparisons in 7 pairs of upper arm arteries and veins from 7 organ donors. SMC: smooth muscle cell, ECM: extracellular matrix, I/I-M: intima/intima-media. * *p* < 0.05, *** *p* < 0.001 by paired *t*-tests. (**D**) Uniform manifold approximation and projection (UMAP) plots of 14,360 cells from 3 arteries and 15,355 cells from 3 veins. (**E**) Relative cell proportions per sample.

**Figure 2 cells-13-00793-f002:**
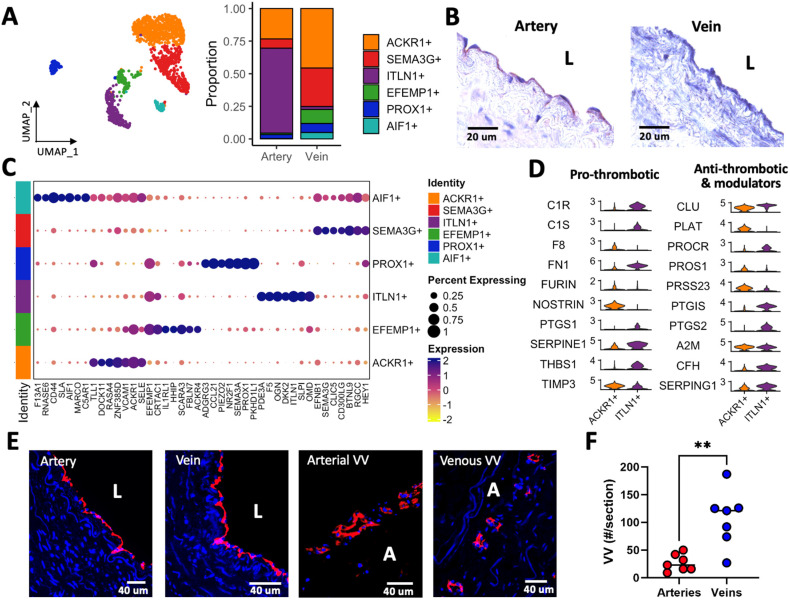
Endothelial cell (EC) characteristics in upper arm arteries and veins: (**A**) Focused UMAP of EC subtypes and mean cell proportions in 3 arteries and 3 veins. Subcluster identities are as follows: *ACKR1*+, venous/venular; *SEMA3G*+, arteriolar/capillary; *ITLN1*+, arterial; *EFEMP1*+, valvular-like; *PROX1*+, lymphatic; *AIF1*+, inflammatory state. (**B**) Representative IHC staining of intelectin 1 (ITLN1) in upper arm arteries and veins. L: lumen. (**C**) Dot plot representation of expression markers for EC subpopulations. Dark purple dots indicate the highest expression levels, while the size of the dot represents the proportion of cells within each subcluster expressing the gene as shown in the legend. (**D**) Hemostatic genes differentially expressed between venous *ACKR1*+ and arterial *ITLN1*+ ECs. (**E**) CD31 staining of arteries and veins showing the main lumen (L) and vasa vasorum (VV) in the adventitia (A). (**F**) Quantification of VV demonstrates increased vascularization in veins vs. arteries (*n* = 7 independent donors per group). ** *p* < 0.01 by *t*-test with Welch’s correction.

**Figure 3 cells-13-00793-f003:**
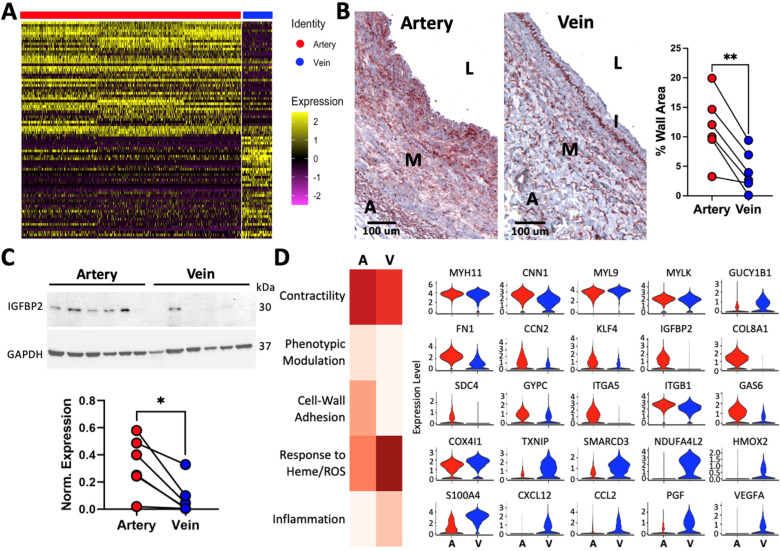
Phenotypic differences between contractile smooth muscle cells (SMCs) from upper arm arteries and veins: (**A**) Heatmap of single-cell expression data for the top differentially expressed genes (DEGs) between contractile SMCs from 3 arteries and 3 veins. Genes are shown in rows and cells in columns. (**B**) Representative IHC staining and quantification of collagen type VIII (COL8A1) in 6 pairs of arteries and veins from 6 donors. L: lumen, I: intima, M: media, A: adventitia. ** *p* < 0.01 by paired *t*-test. (**C**) Quantification of IGFBP2 in 6 pairs of arteries and veins by Western blot. Protein levels were normalized with respect to GAPDH. * *p* < 0.05 by paired *t*-test. (**D**) Heatmap of functional gene signature scores in contractile SMCs from arteries and veins. Representative DEGs corresponding to each score are shown to the right. A: artery, V: vein.

**Figure 4 cells-13-00793-f004:**
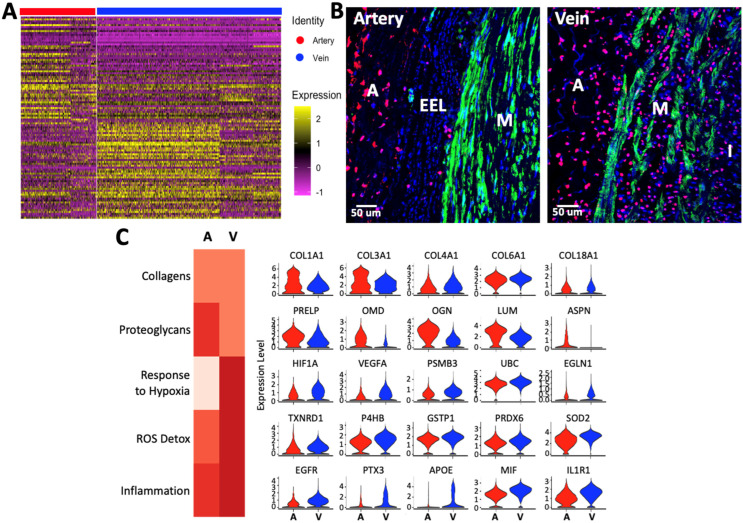
Phenotypic differences between fibroblasts from upper arm arteries and veins: (**A**) Heatmap of single-cell expression data for the top differentially expressed genes (DEGs) between fibroblasts from 3 arteries and 3 veins. Genes are shown in rows and cells in columns. (**B**) Representative IF staining of the SMC and fibroblast markers CNN1 (green) and PDGFRA (red), respectively, in arteries and veins. I: intima, M: media, A: adventitia, EEL: external elastic lamina. (**C**) Heatmap of functional gene signature scores in fibroblasts from arteries and veins. Representative DEGs corresponding to each score are shown to the right. A: artery, V: vein.

**Figure 5 cells-13-00793-f005:**
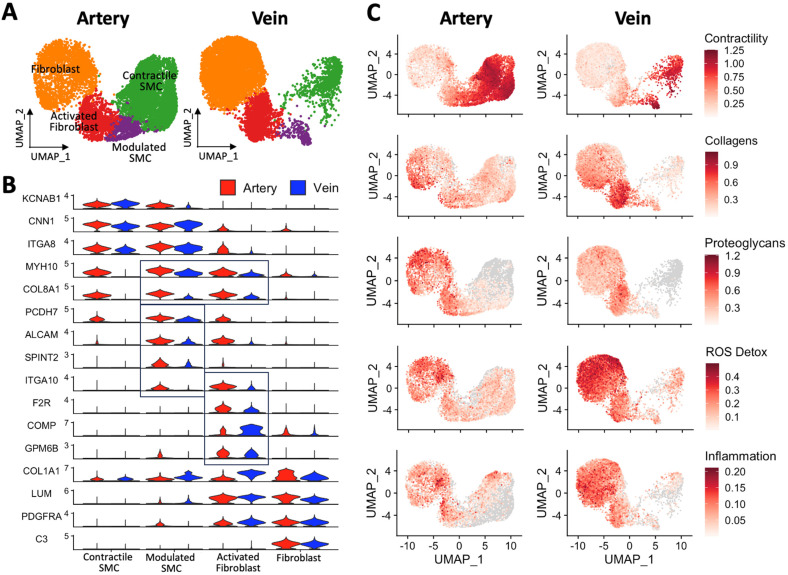
Modulated smooth muscle cells (SMCs) and activated fibroblasts in upper arm arteries and veins: (**A**) Focused UMAP of contractile SMCs, modulated SMCs, activated fibroblasts, and quiescent fibroblasts by vessel type. (**B**) Expression markers defining modulated SMCs and activated fibroblasts in arteries and veins. (**C**) Featured maps of gene signature scores illustrating functional differences between SMC and fibroblast populations from arteries and veins.

**Figure 6 cells-13-00793-f006:**
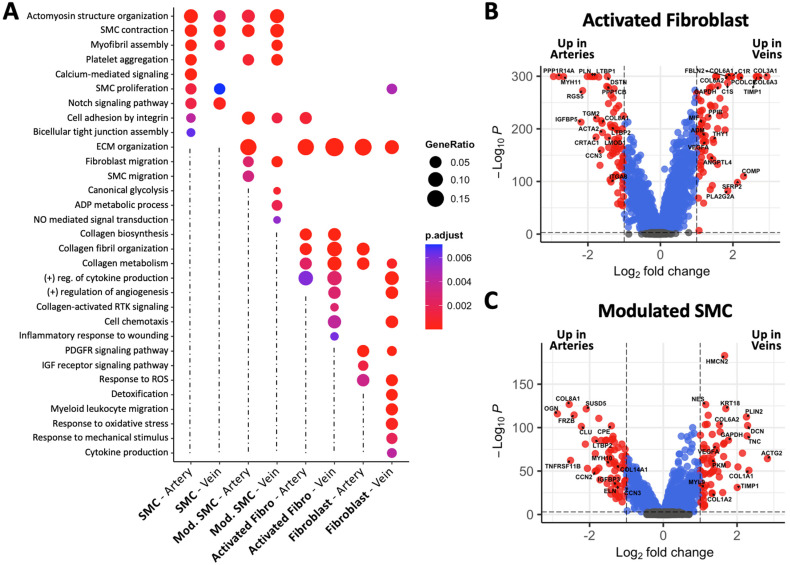
Predicted functional profiles of smooth muscle cells (SMCs) and fibroblast populations from upper arm arteries and veins. (**A**) Gene ontology (GO) over-representation analysis of biological processes in contractile SMCs, modulated SMCs, activated fibroblasts, and fibroblasts from arteries and veins. (**B**,**C**) Volcano plots of differentially expressed genes between activated fibroblasts (**B**) and modulated SMCs (**C**). Expression differences were considered significant (red dots) if log_2_FC > 1 and p.adj < 0.01.

**Figure 7 cells-13-00793-f007:**
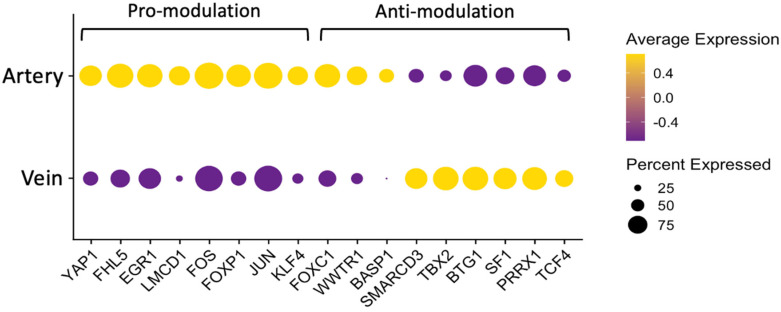
Increased expression of promodulatory transcription factors (TFs) in arterial contractile smooth muscle cells (SMC). Dot plot representation of differentially expressed TFs in arterial and venous contractile SMCs from 3 arteries and 3 veins. Transcription factors were classified as promodulation or antimodulation as reported in the literature. Yellow and purple dots indicate upregulation and downregulation, respectively. The size of the dot represents the percentage of contractile SMCs expressing the gene as the legend on the right indicates.

**Figure 8 cells-13-00793-f008:**
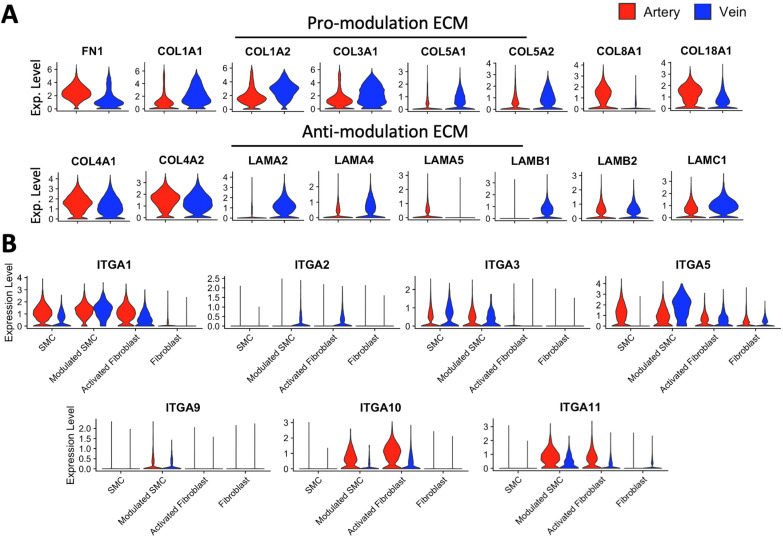
Gene expression differences in extracellular matrix (ECM) genes and receptors between arteries and veins: (**A**) Violin plot representation of normalized expression levels of ECM genes in pooled contractile SMCs, modulated SMCs, activated fibroblasts, and fibroblasts from 3 arteries and 3 veins. ECM components were classified as promodulation or antimodulation as reported in the literature. (**B**) Expression of integrin receptors for collagen (*ITGA1*, *ITGA2*, *ITGA10*, *ITGA11*), laminin (*ITGA3*), fibronectin (*ITGA5*), and osteopontin (*ITGA9*) in contractile SMCs, modulated SMCs, activated fibroblasts, and fibroblasts from arteries and veins.

**Figure 9 cells-13-00793-f009:**
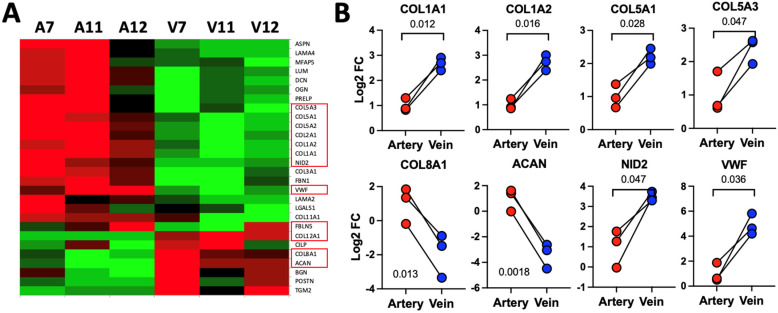
Extracellular matrix (ECM) protein composition in arteries and veins: (**A**) Heatmap of main protein expression differences in 3 pairs of arteries and veins analyzed using ECM-enriched proteomics. Green and red mean upregulated and downregulated, respectively. Only proteins with *p*-value ≤ 0.1 in paired *t*-tests were included in the heatmap. Names in red boxes indicate proteins with significant differences in log_2_FC > |1| and *p* < 0.05. (**B**) Examples of differentially expressed proteins between arteries and veins. Values are presented as log_2_FC of normalized protein abundance. Groups were compared by paired *t*-tests.

**Table 1 cells-13-00793-t001:** Proportions of cell populations in upper arm arteries and veins by single-cell RNA sequencing.

Cell Population	Proportion–Artery (Mean ± SD)	Proportion–Vein (Mean ± SD)	Proportion Ratio (Artery/Vein)	*p*-Value
Endothelial Cell	0.024 ± 0.016	0.096 ± 0.025	0.25	0.021
Pericyte	0.006 ± 0.003	0.027 ± 0.027	0.21	0.045
Contractile SMC	0.510 ± 0.103	0.065 ± 0.046	7.81	0.00016
Modulated SMC	0.079 ± 0.031	0.023 ± 0.013	3.45	0.056
Activated Fibroblast	0.106 ± 0.088	0.103 ± 0.101	1.02	0.97
Fibroblast	0.157 ± 0.096	0.363 ± 0.193	0.43	0.079
Monocyte/Macrophage	0.085 ± 0.063	0.271 ± 0.149	0.32	0.035
NK/T Cell	0.034 ± 0.037	0.053 ± 0.040	0.63	0.37

Cell proportions were compared using the propeller method. Minor cell populations with total cell counts <250 per cluster among all samples analyzed were not considered in the comparison. These included B cells (70 cells), mast cells (204), neutrophils (91), and Schwann cells (95).

**Table 2 cells-13-00793-t002:** Proportions of endothelial cell subpopulations in arteries and veins by single-cell RNA sequencing.

Cell Population	Proportion–Artery (Mean ± SD)	Proportion–Vein (Mean ± SD)	Proportion Ratio (Artery/Vein)	*p*-Value
ITLN1+ (Arterial)	0.542 ± 0.257	0.022 ± 0.012	24.98	<0.0001
ACKR1+ (Venous/venular)	0.332 ± 0.205	0.464 ± 0.102	0.72	0.30
SEMA3G+ (Arteriolar/Cap.)	0.061 ± 0.025	0.287 ± 0.155	0.21	0.018
EFEMP1+ (Valvular-like)	0.020 ± 0.018	0.105 ± 0.109	0.19	0.043
PROX1+ (Lymphatic)	0.043 ± 0.036	0.073 ± 0.050	0.59	0.45
AIF1+ (Inflammatory)	0.001 ± 0.002	0.049 ± 0.009	0.03	0.010

Cell proportions were compared using the propeller method.

**Table 3 cells-13-00793-t003:** Selected expression differences among endothelial cell subpopulations in upper arm arteries and veins.

Symbol	Gene Name	Classification	Upregulated in	Predicted Function
ACKR1	Atypical chemokine receptor 1	EC marker	Veins/venules	Chemotaxis
SEMA3G	Semaphorin 3G	EC marker	Arterioles	EC migration regulation
ITLN1	Intelectin 1	EC marker	Arteries	Carbohydrate binding
EFEMP1	EGF containing fibulin extracellular matrix protein 1	EC marker	Valvular ECs	ECM glycoprotein
PROX1	Prospero homeobox 1	EC marker	Lymphatic ECs	Transcription factor
AIF1	Allograft inflammatory factor 1	EC marker	Inflammatory ECs	
ICAM1	Intercellular adhesion molecule 1	Adhesion	Veins/venules	Leukocyte adhesion
NCAM1	Neural cell adhesion molecule 1	Adhesion	Arteries	Leukocyte adhesion
VCAM1	Vascular cell adhesion molecule 1	Adhesion	Veins/venules	Leukocyte adhesion
SELE	Selectin E	Adhesion	Veins/venules	Leukocyte adhesion
SELL	Selectin L	Adhesion	Arteries	Leukocyte adhesion
SELP	Selectin P	Adhesion	Arteries	Leukocyte adhesion
C1R	Complement C1r	Complement	Arteries	Complement cascade
C1S	Complement C1s	Complement	Arteries	Complement cascade
CFH	Complement factor H	Complement	Arteries	Complement cascade
NOSTRIN	Nitric oxide synthase trafficking inducer	Adapter	Veins/venules	Reduced eNOS activity
PLAT	Tissue-type plasminogen activator	Protease	Veins/venules	Fibrinolysis
PTGIS	Prostaglandin I2 synthase	Enzyme	Arteries	PG biosynthesis
PTGS1	Prostaglandin endoperoxide synthase 1	Enzyme	Arteries	PG biosynthesis
PTGS2	Prostaglandin endoperoxide synthase 2	Enzyme	Arteries	PG biosynthesis
SERPINE1	Serpin family E member 1	Inhibitor	Arteries	Fibrinolysis inhibitor
SERPING1	Serpin family G member 1	Inhibitor	Arteries	Complement inhibitor

**Table 4 cells-13-00793-t004:** Expression markers and selected DEGs in smooth muscle cells and fibroblasts between arteries and veins.

Symbol	Gene Name	Classification	Upregulated in	Predicted Function
**Smooth muscle cells (SMC)**				
ACTA2	Actin alpha 2, smooth muscle	SMC marker		Contractility
TAGLN	Transgelin	SMC marker		Contractility
MYH11	Myosin heavy chain 11	SMC marker		Contractility
MYL9	Myosin light chain 9	SMC marker		Contractility regulation
MYLK	Myosin light chain kinase	SMC marker		Contractility
CNN1	Calponin 1	SMC marker	Arteries	Contractility regulation
GUCY1A1	Guanylate cyclase 1 soluble subunit alpha 1	Enzyme	Veins	Vasodilation
GUCY1B1	Guanylate cyclase 1 soluble subunit beta 1	Enzyme	Veins	Vasodilation
ITGA1	Integrin subunit alpha 1	Integrin	Arteries	Collagen/laminin receptor
ITGA5	Integrin subunit alpha 5	Integrin	Arteries	Fibronectin receptor
SORBS1	Sorbin and SH3 domain containing 1	Integrin complex	Arteries	Cell adhesion
VCL	Vinculin	Integrin complex	Arteries	Cell adhesion
FN1	Fibronectin 1	ECM	Arteries	ECM remodeling
SDC4	Syndecan 4	Proteoglycan	Arteries	Fibronectin binding
FMOD	Fibromodulin	Proteoglycan	Arteries	ECM remodeling
COL8A1	Collagen type VIII alpha 1 chain	Collagen	Arteries	ECM remodeling
KLF4	Kruppel-like factor 4	Transcription factor	Arteries	Phenotypic modulation
CCN2	Cellular communication network factor 2	Growth factor	Arteries	Phenotypic modulation
IGFBP2	Insulin-like growth factor binding protein 2	Growth factor	Arteries	Phenotypic modulation
TNFRSF11B	TNF receptor superfam. member 11b	Decoy receptor	Arteries	Phenotypic modulation
COX4I1	Cytochrome c oxidase subunit 4I1	Enzyme	Veins	Cellular respiration
NDUFA4L2	NDUFA4, mitochondrial-complex-associated like 2	Enzyme	Veins	Cellular respiration regulation
HMOX2	Heme oxygenase 2	Enzyme	Veins	Hemoglobin metabolism
CCL2	C-C motif chemokine ligand 2	Chemokine	Veins	Chemotaxis
CXCL12	CXC motif chemokine ligand 12	Chemokine	Veins	Chemotaxis
PGF	Placental growth factor	Growth factor	Veins	Angiogenesis
VEGFA	Vascular endothelial growth factor A	Growth factor	Veins	Angiogenesis
S100A4	S100 calcium-binding protein A4	Secreted factor	Veins	Inflammation
**Fibroblasts**				
PDGFRA	Platelet-derived growth factor receptor alpha	Fibro marker		Proliferation
DCN	Decorin	Fibro marker		ECM remodeling
C3	Complement 3	Fibro marker		Complement signaling
CFD	Complement factor D	Fibro marker		Complement signaling
LUM	Lumican	Fibro marker	Arteries	ECM remodeling
PRELP	Proline and arginine-rich end leucine-rich repeat protein	Proteoglycan	Arteries	ECM remodeling
OMD	Osteomodulin	Proteoglycan	Arteries	ECM remodeling
OGN	Osteoglycin	Proteoglycan	Arteries	ECM remodeling
ASPN	Asporin	Proteoglycan	Arteries	ECM remodeling
HIF1A	Hypoxia-inducible factor 1 subunit alpha	Transcription factor	Veins	Hypoxia response
EGLN1	Egl-9 fam. Hypoxia-inducible factor 1	Oxygen sensor	Veins	Hypoxia response
PSMB3	Proteasome 20S subunit beta 3	Proteolytic factor	Veins	Hypoxia response
UBC	Ubiquitin C	Proteolytic factor	Veins	Hypoxia response
VEGFA	Vascular endothelial growth factor A	Growth factor	Veins	Hypoxia response
P4HB	Prolyl 4-hydroxylase subunit beta	Enzyme	Veins	Redox homeostasis
PRDX6	Peroxiredoxin 6	Enzyme	Veins	Redox homeostasis
TXNRD1	Thioredoxin reductase 1	Enzyme	Veins	Redox homeostasis
SOD2	Superoxide dismutase 2	Enzyme	Veins	Redox homeostasis
ADM	Adrenomedullin	Hormone	Veins	Vasodilation
CXCL2	CXC motif chemokine ligand 2	Chemokine	Veins	Chemotaxis
CXCL3	CXC motif chemokine ligand 3	Chemokine	Veins	Chemotaxis
CXCL12	CXC motif chemokine ligand 12	Chemokine	Veins	Chemotaxis
EGFR	Epidermal growth factor receptor	Receptor	Veins	Proliferation
IL1R1	Interleukin 1 receptor type 1	Receptor	Veins	Inflammation
MIF	Macrophage migration inhibitory factor	Lymphokine	Veins	Inflammation

## Data Availability

Data have been deposited in the Gene Expression Omnibus (GEO), accession numbers GSE266682 and GSE250469 (for single-cell RNA sequencing), and GSE266681 and GSE233264 (for bulk RNA sequencing).

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
