# Peer review of "Single-Cell Analyses Offer Insights into the Different Remodeling Programs of Arteries and Veins"

_cells, 2024, doi:10.3390/cells13100793_

Round 1

Reviewer 1 Report

Comments and Suggestions for Authors

The study present sound results about single cells analysis comparing arteries and veins. It presents different methodologies to determine the expression of characteristic markers and the quantification of different cell types in each vessel. 

The article complexity turns the its reading difficult since several figures that are in the supplemental data are necessary to completely understand the authors conclusions. The main suggestion is to overlook all figures from data and supplemental data and try to determine the most important figures necessary for the reading without going back and fourth to the supplemental data. E.g. the studied genes could de showed in a table, instead in the text like in the paragraph from line 388 to 408. 

Author Response

The study present sound results about single cells analysis comparing arteries and veins. It presents different methodologies to determine the expression of characteristic markers and the quantification of different cell types in each vessel.

Q1. The article complexity turns the reading difficult since several figures that are in the supplemental data are necessary to completely understand the authors conclusions. The main suggestion is to overlook all figures from data and supplemental data and try to determine the most important figures necessary for the reading without going back and forth to the supplemental data. E.g. the studied genes could be showed in a table, instead in the text like in the paragraph from line 388 to 408.

A1. We appreciate the reviewer’s comments and insightful suggestions. We have reduced the number of supplementary figures from 13 to 8, either by removing figures with redundant information or combining figure panels with new figures in the main manuscript (new Figure 2, modified Figure 8). We have also moved a supplementary table to the main text (Table 1). To improve clarity, we have prepared two summary tables (Tables 3 and 4) with gene names, classification, direction of regulation, and function of the main expression markers and differentially expressed genes in ECs, SMCs, and fibroblasts. While we could not include all functional relationships discussed in the text due to space limitations, many of the genes discussed in the myofibroblast section are also mentioned for the SMC and fibroblast clusters in Table 4.

Reviewer 2 Report

Comments and Suggestions for Authors

This is an interesting manuscript that aims to contrast cell populations and gene expression between human arteries and veins with the ultimate goal of advancing our understanding of the differences in remodeling potential and mechanisms between these vessel types.  The authors utilize single-cell and bulk RNA sequencing, histology/immunocytochemistry, flow cytometry and proteomics to provide a thorough and insightful analysis of this question.  The authors present some relatively complex data in a relatively clear manner.  I have several questions/suggestions related primarily to data presentation:

-          -  What is the difference between the histological images in Figures S2 and 1A and 1B?  These seem redundant and Figure S2 could be eliminated unless there is a rationale for including all of these images.

-           - The authors define figure labels in the figure legends and this is very helpful.  The “A” and “L” used in Figure S3A should be defined in the legend.

-          -  It would be helpful to include standard error or standard deviation in Table S2.

-          -  It might be beneficial to include common names for the genes that are discussed in the text either in a separate table or by incorporating them into one of the existing spreadsheets. 

-          -  Is Figure S5 necessary as similar data is already presented in Figure 3B?

-           - The authors include several comments regarding relevance of the data in the Results section (lines 262 – 264, for instance) that would be more appropriate for the Discussion.

-           - The authors indicate that COL8A1protein is more abundant in the arterial wall; however, this was not quantified.  The IHC appears to indicate this, but image analysis should be performed to support this statement.

Author Response

This is an interesting manuscript that aims to contrast cell populations and gene expression between human arteries and veins with the ultimate goal of advancing our understanding of the differences in remodeling potential and mechanisms between these vessel types.  The authors utilize single-cell and bulk RNA sequencing, histology/immunocytochemistry, flow cytometry and proteomics to provide a thorough and insightful analysis of this question.  The authors present some relatively complex data in a relatively clear manner.  I have several questions/suggestions related primarily to data presentation:

Q1. What is the difference between the histological images in Figures S2 and 1A and 1B? These seem redundant and Figure S2 could be eliminated unless there is a rationale for including all of these images.

A1. We agree. We thank the Reviewer for pointing this out. Following the recommendations of this Reviewer and Reviewer 1, we have removed any redundant supplementary figures (including Figure S2), reducing the total number of supplementary figures from 13 to 8.   

Q2. The authors define figure labels in the figure legends and this is very helpful.  The “A” and “L” used in Figure S3A should be defined in the legend.

A2. We completely agree. We know indicate that L refers to the lumen and A to the adventitia in the figure legend. In addition, following the suggestion of Reviewer 3, we have expanded the analyses of EC populations in arteries and veins and renumbered this figure as Figure 2 in the main manuscript.

Q3. It would be helpful to include standard error or standard deviation in Table S2.

A3. Fully agree. We have included this information in the table (now Table 1) and the new Table 2 for EC populations as well.

Q4. It might be beneficial to include common names for the genes that are discussed in the text either in a separate table or by incorporating them into one of the existing spreadsheets.

A4. Agree. Following the recommendations of this Reviewer and Reviewer 1, we have prepared two summary tables with the main expression markers and differentially expressed genes discussed in the text (Tables 3 and 4).

Q5. Is Figure S5 necessary as similar data is already presented in Figure 3B?

A5. Agree. We have removed this figure as indicated in A1 above.

Q6. The authors include several comments regarding relevance of the data in the Results section (lines 262 – 264, for instance) that would be more appropriate for the Discussion.

A6. We agree with the Reviewer. However, we find that it the style of transcriptomic articles to include conceptual information about genes in the Results to make the information more amenable, improve flow, and establish functional relationships.

Q7. The authors indicate that COL8A1 protein is more abundant in the arterial wall; however, this was not quantified.  The IHC appears to indicate this, but image analysis should be performed to support this statement.

A7. We completely agree. We apologize for this omission. Figure 2 now includes a quantification of COL8A1+ area in 6 pairs of arteries and veins. The quantification corroborates the RNA expression differences and the findings from the proteomics analysis (Figure 9).

Reviewer 3 Report

Comments and Suggestions for Authors

The manuscript written by Miguel G. Rojas et al. talks about the various gene expression profiles in human arteries and veins by single-cell analysis. The manuscript is well written, with detailed information on changes in gene expression patterns in SMCs and fibroblasts. Critical comments;

1.     Decreased needs to be indicated following 30 in the first line of Section 2.1. Furthermore, it's unclear what led to these patients' deaths. For instance, variations in the gene expression of the cells produced from these organs may influence the cause of death.

  1. The authors mostly targeted atherosclerosis disease in the entire manuscript; however, I don’t see any coagulation-related gene analysis in veins and arteries. Coagulation is closely associated with the process of atherosclerosis.
  2. The authors have mentioned in the discussion that “Lastly, the increased inflammatory milieu in veins vs. arteries by virtue of chemokine-expressed 536 pressing ECs, SMCs, myofibroblastic cells, and fibroblasts” 

However, the gene expression profile of endothelial cells is discussed too little. Endothelial cells are in direct contact with the blood, and considering the difference in flow rate between veins and arteries, there will be differences in gene expression. Did the authors analyze the gene expression in endothelial cells similar to what they did in SMCs and fibroblasts?

Comments on the Quality of English Language

N/A

Author Response

The manuscript written by Miguel G. Rojas et al. talks about the various gene expression profiles in human arteries and veins by single-cell analysis. The manuscript is well written, with detailed information on changes in gene expression patterns in SMCs and fibroblasts.

Q1. Deceased needs to be indicated following 30 in the first line of Section 2.1. Furthermore, it's unclear what led to these patients' deaths. For instance, variations in the gene expression of the cells produced from these organs may influence the cause of death.

A1. Thank you very much for pointing out these omissions. We modified the sentence as suggested. Table S1 also includes a new column with the causes of death for all tissue donors. In light of this Reviewer’s comment, we have additionally included a limitations statement (lines 604-606) acknowledging this possibility.

Q2. The authors mostly targeted atherosclerosis disease in the entire manuscript; however, I don’t see any coagulation-related gene analysis in veins and arteries. Coagulation is closely associated with the process of atherosclerosis.

A2. We fully agree. We have now included a new Figure 2 and Results section expanding the analysis of ECs from arteries and veins, including an evaluation of RNA expression differences in coagulation cascade genes. The results suggests that different regulatory mechanisms control hemostasis in these types of vessels.

Q3. The authors have mentioned in the discussion that “Lastly, the increased inflammatory milieu in veins vs. arteries by virtue of chemokine-expressed 536 pressing ECs, SMCs, myofibroblastic cells, and fibroblasts” However, the gene expression profile of endothelial cells is discussed too little. Endothelial cells are in direct contact with the blood, and considering the difference in flow rate between veins and arteries, there will be differences in gene expression. Did the authors analyze the gene expression in endothelial cells similar to what they did in SMCs and fibroblasts?

A3. Please see above answer A2.